# The AE4 transporter mediates kidney acid-base sensing

H. Vitzthum[1], M. Koch[1], L. Eckermann[1], S. L. Svendsen[2], P. Berg [2], C. A. Hübner [3], C. A. Wagner [4,5], J. Leipziger[2], C. Meyer-Schwesinger [1] & H. Ehmke [1,6] ✉

The kidney plays a key role in the correction of systemic acid-base imbalances. Central for this regulation are the intercalated cells in the distal nephron, which secrete acid or base into the urine. How these cells sense acid-base disturbances is a long-standing question. Intercalated cells exclusively express the Na$^+$-dependent Cl$^-$/HCO$_3^-$ exchanger AE4 (*Slc4a9*). Here we show that AE4-deficient mice exhibit a major dysregulation of acid-base balance. By combining molecular, imaging, biochemical and integrative approaches, we demonstrate that AE4-deficient mice are unable to sense and appropriately correct metabolic alkalosis and acidosis. Mechanistically, a lack of adaptive base secretion via the Cl$^-$/HCO$_3^-$ exchanger pendrin (*Slc26a4*) is the key cellular cause of this derailment. Our findings identify AE4 as an essential part of the renal sensing mechanism for changes in acid-base status.

Maintenance of blood pH in its narrow physiological range is essential for cellular and organismal health. This is achieved by a collaborative effort of the kidney and respiratory system. Here, kidney tubular epithelial cells balance systemic acid-base status by regulating the recovery, de novo synthesis, and secretion of bicarbonate (HCO$_3^-$), the most important base in the extracellular compartment on the one hand, and on the other hand by regulating the secretion of acid in the form of hydrogen ions (H$^+$)[1]. Failure to maintain systemic pH within limits compatible with normal cell and organ function causes disease and constitutes a major complication of other underlying diseases[2]. Acid-base disturbances are an independent risk factor for morbidity and mortality in the setting of various conditions[3,4].

How renal tubular cells sense and respond to acid-base changes remains poorly defined, and the common view is that a single master regulator does not exist but that multiple systems and molecules regulate specific components[5,6]. The intercalated cells (ICs) of the kidney collecting duct system play a decisive role for the kidneys' ability to regulate acid-base homeostasis. α-ICs secrete acid in form of H$^+$ through the action of a H$^+$-ATPase and H$^+$/K$^+$-ATPases[1,7,8]. β-ICs are unique in their capability to secrete base in form of HCO$_3^-$ through the luminal Cl$^-$/HCO$_3^-$ exchanger pendrin (*Slc26a4*). Intracellular pH in β-IC is maintained by H$^+$ extrusion across the basolateral membrane via H$^+$-ATPases[9]. By this, collecting duct HCO$_3^-$ secretion is linked to Cl$^-$ and H$^+$ reabsorption. Depending on systemic acid-base status, β-ICs regulate pendrin abundance and subcellular localization to either enhance or decrease HCO$_3^-$ secretion. During metabolic alkalosis, pendrin is rapidly trafficked to the apical membrane of β-ICs, hence increasing HCO$_3^-$ secretion[10,11] while in metabolic acidosis the abundance and apical localization of pendrin is reduced[10–12].

## Results

### Localization of AE4 in different species

In rodents, intercalated cells (ICs) abundantly and uniquely express the membrane protein AE4 (*Slc4a9*) on the mRNA[13,14] and protein level[12,15]. Immunolocalization in murine, porcine, and human kidney sections of AE4 together with the principal cell marker aquaporin 2 (APQ2), and the marker for β-ICs pendrin confirmed an exclusive basolateral (blood-facing) expression conserved among species (Fig. 1a).

[1]Center for Experimental Medicine, Institute of Cellular and Integrative Physiology, University Medical Center Hamburg-Eppendorf, Hamburg, Germany. [2]Department of Biomedicine, Physiology, Health, Aarhus University, Aarhus, Denmark. [3]Institute of Human Genetics, University Hospital Jena, Friedrich Schiller University, Jena, Germany. [4]Institute of Physiology, University of Zurich, Zurich, Switzerland. [5]National Center of Competence in Research NCCR Kidney.CH, Zurich, Switzerland. [6]German Center for Cardiovascular Research, Partner Site Hamburg/Kiel/Lübeck, Hamburg, Germany. ✉e-mail: ehmke@uke.de

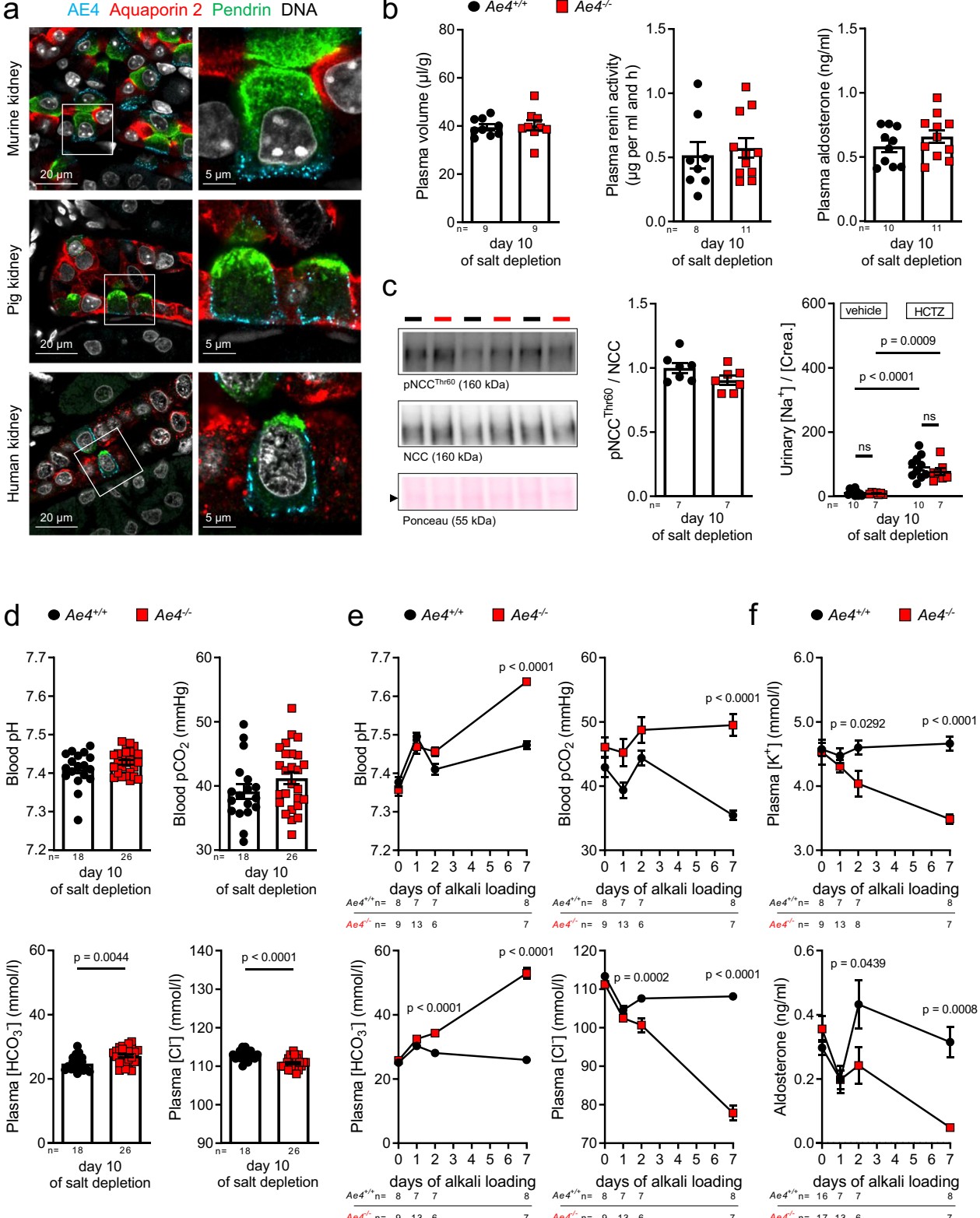

## AE4 is not involved in maintaining salt-water balance

The physiological function of AE4 in ICs in vivo is unknown, even though its expression was documented more than 20 years ago[16]. AE4 is an electroneutral monovalent cation-dependent $Cl^-/HCO_3^-$ exchanger[17] and is currently thought to be critically involved in blood-volume regulation by mediating $Na^+$-reabsorption in β-ICs[18,19]. How-ever, we could not confirm this function in in vivo studies in mice with loss of AE4 ($Ae4^{-/-}$). Specifically, $Ae4^{-/-}$ mice did not exhibit impaired blood-volume regulation under dietary salt-depletion, which would be expected if AE4 was necessary for a substantial NaCl-reabsorption. In both $Ae4^{-/-}$ mice and wild-type littermates ($Ae4^{+/+}$), a dietary salt-depletion resulted in a comparable decrease of urinary $Na^+$ and $Cl^-$ excretion (Supplementary Fig. 1a). Consequently, plasma volume (Fig. 1b), hematocrit and blood urea nitrogen (Supplementary Fig. 1b)

**Fig. 1 | Basolateral AE4 of β-intercalated cells is not required for systemic sodium balance but for acid-base homeostasis. a** Representative confocal images of AE4 (light blue), AQP2 (red), and pendrin (green) in murine, porcine and human kidney demonstrating basal and lateral localization of AE4 in cortical pendrin-positive cells. **b–d** $Ae4^{-/-}$ and wild-type littermates ($Ae4^{+/+}$) were challenged for 10 days with a salt-depleted diet. **b** Plasma volume measured by Evans blue dilution method, plasma renin activity and aldosterone concentration measured by ELISA ($n = 8$-11 animals per genotype, two-tailed Student's $t$-test). **c** Representative immunoblots of total renal sodium chloride cotransporter NCC and phosphorylated (p)NCC protein levels, ponceau red staining indicates equal loading. Graphs exhibit densitometric quantification of protein levels normalized to ponceau red staining, relative levels to $Ae4^{+/+}$ are plotted ($n = 7$ animals per genotype, two-tailed Student's t-test). The effect of the NCC blocker hydrochlorothiazide (HCTZ, 10 mg/kg) on urinary $Na^+$ was not different between both genotypes ($n = 7$–10 animals per genotype, ns: not significant $p > 0.05$, two-way ANOVA followed by Bonferroni's multiple comparisons test). **d** AE4 deletion induces hypochloremic alkalosis during salt-depletion. Graphs exhibit blood pH, blood pCO$_2$, plasma [HCO$_3^-$], and plasma [Cl$^-$] ($n = 18$–26 animals per parameter and genotype, two-tailed Student's $t$-test). **e, f** $Ae4^{+/+}$ and $Ae4^{-/-}$ mice were alkali-loaded for 7 days under salt-depletion (low salt diet with 230 mM NaHCO$_3$ added to the drinking water). Before the alkali-loading (day 0) mice received normal diet. Graphs show **e** blood pH, plasma [HCO$_3^-$], blood pCO$_2$, and plasma [Cl$^-$] and **f** plasma [K$^+$] and [aldosterone] of $Ae4^{+/+}$ and $Ae4^{-/-}$ littermates without and after 1, 2, and 7 days of alkali-loading ($n = 6$–17 animals per genotype and day of alkali-loading, two-way ANOVA followed by Bonferroni's multiple comparisons test for $Ae4^{+/+}$ vs $Ae4^{-/-}$). All data are presented as mean ± SEM. In **b–d** each point or square denotes one animal. Source data are provided as a Source Data file.

and the activity of the salt-sparing renin-angiotensin-aldosterone system (RAAS) (Fig. 1b) did not differ between genotypes. Consistently, the abundance and activity of compensatory renal Na$^+$-reabsorption pathways, such as the Na$^+$/Cl$^-$ cotransporter NCC (Fig. 1c, Supplementary Fig. 2a), as well as the epithelial Na$^+$ channel ENaC (Supplementary Fig. 2b), did not differ between genotypes. The natriuretic response to the NCC blocking diuretic hydrochlorothiazide (HCTZ) was comparable between $Ae4^{+/+}$ and $Ae4^{-/-}$ mice (Fig. 1c) indicating the absence of any compensatory upregulation of salt reabsorption in the distal tubule of $Ae4^{-/-}$ mice, as present in other salt-losing conditions[20].

### Role of AE4 in the maintenance of acid-base balance

Unexpectedly, $Ae4^{-/-}$ mice exhibited a perturbation of their acid-base status in the setting of salt-depletion. Here, $Ae4^{-/-}$ but not wild-type littermates exhibited hypochloremic metabolic alkalosis (decreased plasma [Cl$^-$]) with elevated plasma [HCO$_3^-$] (Fig. 1d), and a positive base excess (BE, Supplementary Fig. 3a). This observation prompted us to evaluate whether AE4 was involved in renal acid-base homeostasis. We therefore additionally challenged salt-depleted $Ae4^{-/-}$ mice and wild-type littermates with dietary alkali-loading (230 mM NaHCO$_3$, 7 days). This resulted in the development of life-threatening hypochloremic alkalosis in $Ae4^{-/-}$ mice. Initially, upon 1 day of alkali-loading, both wild-type and $Ae4^{-/-}$ mice exhibited an increase of blood pH and plasma [HCO$_3^-$] together with a slightly decreased plasma [Cl$^-$]. While wild-type littermates normalized their acid-base status hereafter, $Ae4^{-/-}$ mice developed a severe hypochloremic alkalosis with a profound secondary respiratory response (increased blood pCO$_2$) (Fig. 1e, Supplementary Fig. 3b). This acid-base disturbance was load-dependent, as $Ae4^{-/-}$ mice receiving higher amounts of NaHCO$_3$ exhibited a more severe hypochloremic alkalosis (Supplementary Fig. 3c). In addition, $Ae4^{-/-}$ mice exhibited a substantial weight loss over the 7-day time course (Supplementary Fig. 3d), accompanied by hypokalemia and decreasing plasma concentrations of the adrenal hormone aldosterone (Fig. 1f). As the release of aldosterone is substantially regulated by plasma [K$^+$], the observed drop of plasma [aldosterone] was the predicted physiologic consequence of the alkalosis-induced hypokalemia in the dietary-challenged $Ae4^{-/-}$ mice. The severe weight loss was in part due to a reduced food intake (Supplementary Fig. 3d) and a profound volume depletion, as indicated by an increased hematocrit, high plasma renin activity, and renal sodium and water loss in the $Ae4^{-/-}$ mice (Supplementary Fig. 3e). The disrupted renal sodium and water handling was most likely the consequence of the derailed physiological status of the $Ae4^{-/-}$ mice after 7 days of alkali-loading. Alkalosis and hypoaldosteronism decreased the abundance of main sodium reabsorption pathways along the nephron (Supplementary Fig. 3f), whereas hypokalemia impaired the renal ability to concentrate the urine[21].

### AE4-deficient mice lack pendrin activation upon alkalosis

As the Cl$^-$/HCO$_3^-$ exchanger pendrin ($Slc26a4$) represents the exclusive apical route for renal HCO$_3^-$ secretion and Cl$^-$ retention, we questioned whether a loss of pendrin activation represents the cellular mechanism leading to the life-threatening condition with hypochloremic alkalosis, massive hypokalemia, and volume loss in alkali-loaded and salt-depleted $Ae4^{-/-}$ mice. Normally, the protein abundance of pendrin is upregulated in metabolic alkalosis to increase HCO$_3^-$ secretion into urine and Cl$^-$ reabsorption into blood[10,22]. Since pendrin is also activated by aldosterone independent of concurrent alkalosis[23–25], we analyzed pendrin regulation on day 1 of alkali-loading when aldosterone levels were identical between genotypes, and on day 7 of alkali-loading when aldosterone levels were suppressed in $Ae4^{-/-}$ mice (Fig. 1f). Our analysis demonstrated that independently of aldosterone levels, $Ae4^{-/-}$ mice fail to activate pendrin. Specifically, wild-type mice showed the expected physiological upregulation of pendrin mRNA and protein abundance compared to unloaded wild-type mice after 1 and 7 days of alkali-loading, whereas $Ae4^{-/-}$ mice failed to adapt pendrin mRNA or protein expression (Fig. 2a, b). In line with previous reports[10,26], alkali-loading induced a pendrin redistribution from the subapical cytosolic region to the apical membrane in wild-type mice, which was absent in $Ae4^{-/-}$ mice (Fig. 2c). Strikingly, after 7 days of loading, $Ae4^{-/-}$ mice not only exhibit a reduced pendrin abundance but also a reduced relative number of pendrin positive cells despite severe alkalosis (Fig. 2d). To assess pendrin activity, we perfused isolated cortical collecting ducts (CCD) and measured the intracellular pH (pH$_i$) of β-ICs[27]. The pendrin transport rate, measured as pH$_i$ change during recovery after a luminal Cl$^-$ removal, was not different between unloaded wild-type and $Ae4^{-/-}$ mice. However, after 1 day of alkali-loading, an increase of pendrin transport rate was evident in wild-type but not in $Ae4^{-/-}$ CCDs (Fig. 2e, Supplementary Fig. 4a). Thus, as expected, enhanced pendrin expression and redistribution to the apical membrane in wild-type mice was paralleled by an augmented pendrin activity and urinary HCO$_3^-$ excretion (Fig. 2f, Supplementary Fig. 4b). This physiological response was entirely absent in alkali-loaded $Ae4^{-/-}$ mice. Taken together, $Ae4^{-/-}$ mice lack the rapid activation of pendrin upon alkali-loading and enter a vicious cycle of alkalosis, hypokalemia, low aldosterone levels, reduction of pendrin, and aggravation of alkalosis (Supplementary Fig. 4c). Importantly, insufficient pendrin stimulation upon alkali-loading was also apparent under normal dietary salt intake in $Ae4^{-/-}$ mice (Supplementary Fig. 5a–d). These data demonstrate, that AE4 is essential for the initial activation of pendrin in renal β-ICs upon alkali-loading.

### AE4-deficient mice fail to reduce pendrin upon acidosis

As $Ae4^{-/-}$ mice failed to adequately sense and respond to alkali-loading, we next asked whether AE4-deficiency also impairs acid sensing. Acid-loading causes a reduction of pendrin expression and a shift of pendrin from the apical membrane to the subapical region[10,12]. To mirror the alkali-loading approach, we challenged salt-depleted $Ae4^{-/-}$ and wild-type littermates with an acid-load for one day (low salt diet with 280 mM NH$_4$Cl). Both genotypes developed a hyperchloremic acidosis, but the acidosis was much more pronounced in $Ae4^{-/-}$ mice.

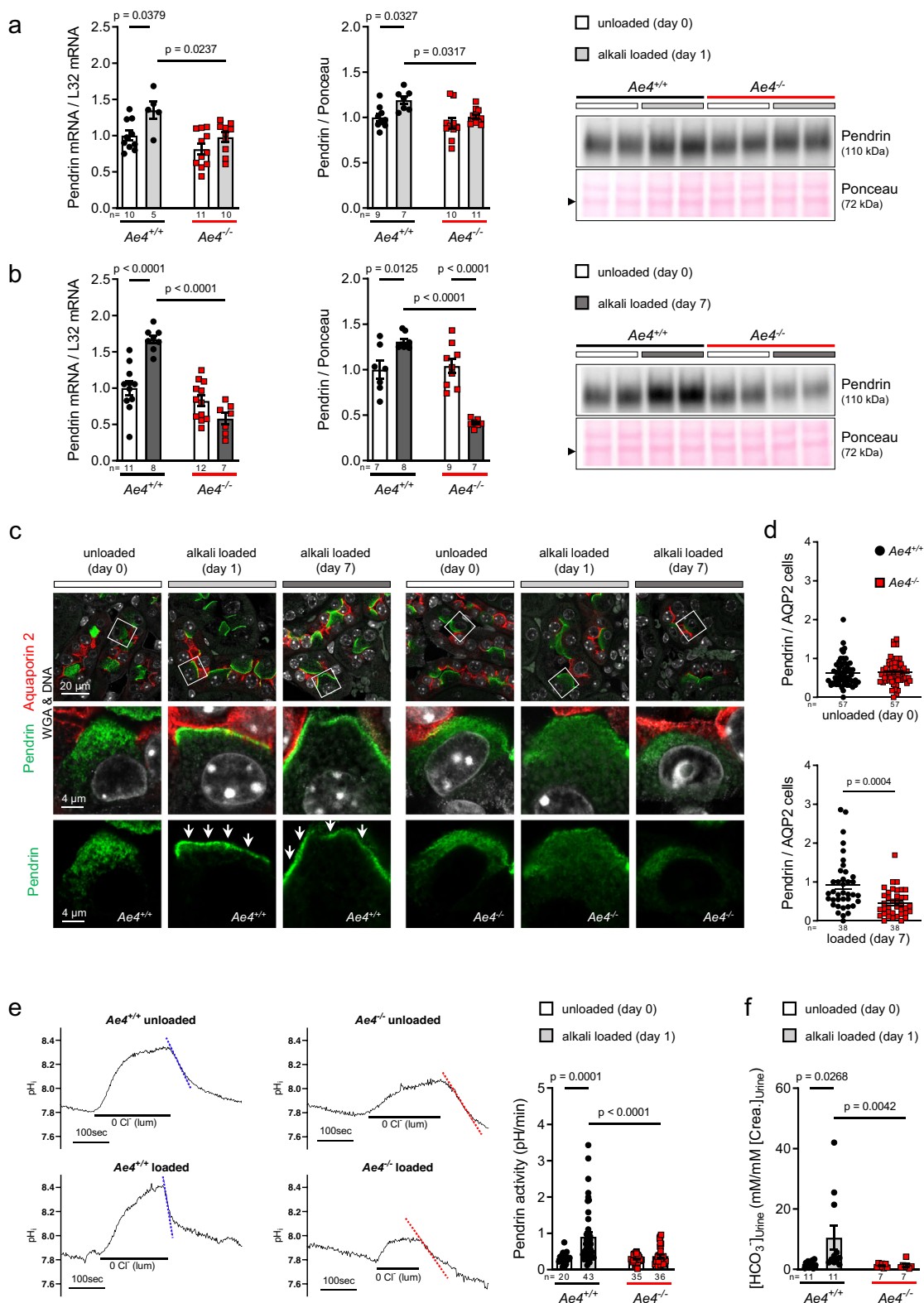

Specifically, blood pH, BE, and plasma [HCO$_3^-$] were significantly lower in $Ae4^{-/-}$ compared to wild-type littermates, whereas plasma [Cl$^-$] was significantly higher (Fig. 3a, Supplementary Fig. 6a). Blood pCO$_2$, plasma [K$^+$], and plasma [aldosterone] (Supplementary Fig. 6b) were not different between genotypes. Analysis of pendrin mRNA and protein expression revealed that only wild-type mice decreased pendrin mRNA and protein abundance upon acid-loading, whereas both

remained unchanged in acid-loaded $Ae4^{-/-}$ mice (Fig. 3b). A typical morphologic feature in acidosis is a reduction of the so-called "pendrin cap" size, which is caused by endocytosis of pendrin from the apical membrane upon acid-loading[12]. While wild-type mice exhibited this reduction of pendrin cap size, this response was entirely absent in $Ae4^{-/-}$ mice (Fig. 3c, d). Upon acid-loading urine pH decreased (Fig. 3e), whereas titratable acid and NH$_4^+$ excretion (Supplementary Fig. 6c)

**Fig. 2 | AE4 is essential for pendrin stimulation in metabolic alkalosis.** *Ae4*⁻/⁻ and wild-type littermates (*Ae4*⁺/⁺) were alkali-loaded for 7 days under salt-depletion (low salt diet with 230 mM NaHCO₃ added to the drinking water). Before alkali-loading (day 0) mice received normal diet. **a** (day 1) and **b** (day 7) show representative immunoblots of pendrin in *Ae4*⁺/⁺ and *Ae4*⁻/⁻ whole kidney lysates. Ponceau red staining served as control for loading. Graphs exhibit densitometric quantification of pendrin mRNA and protein levels (normalized to the levels found in *Ae4*⁺/⁺ mice at day 0, *n* = 5–12 animals each genotype and day of diet, one-way ANOVA followed by Bonferroni's multiple comparisons test). **c** Immunofluorescence staining of pendrin (green) and AQP2 (red) in the kidneys of mice before (day 0) and after 1 or 7 days of alkali-loading. In *Ae4*⁺/⁺ mice, but not in the *Ae4*⁻/⁻ littermates redistribution of pendrin to the apical membrane region (white arrows) upon alkali-loading was observed. After 7 days of loading pendrin staining in *Ae4*⁻/⁻ was reduced. **d** Ratio of pendrin positive cells / AQP2 positive cells in the renal cortex determined for unloaded (day 0) and loaded (day 7) mice (*n* = 38–57 pictures of cortical fields of each genotype and day of diet, two-tailed Mann–Whitney test). Note the relative decrease of pendrin-positive cells in *Ae4*⁻/⁻ mice after 7 days of alkali-loading. **e** Pendrin activity in *Ae4*⁺/⁺ and *Ae4*⁻/⁻ littermates before (day 0) and after alkali-loading (day 1). In unloaded mice pendrin function was not different between mice of the two genotypes. After 1 day of alkali-loading *Ae4*⁺/⁺ mice amplified pendrin activity, whereas no change of pendrin activity was observed in *Ae4*⁻/⁻ mice (*n* = 20–43 cells of each genotype and day of diet, one-way ANOVA followed by Bonferroni's multiple comparisons test). **f** Urine alkalization measured as urinary [HCO₃⁻] normalized to urinary [creatinine] (*n* = 7–11 animals each genotype and day of diet, Kruskal–Wallis followed by Dunn's multiple comparisons test). Urine alkalization induced by alkali-loading was absent in *Ae4*⁻/⁻ mice. All data are presented as mean ± SEM. In **a**, **b**, **f** each point or square denotes one animal. Source data are provided as a Source Data file.

increased in mice of both genotypes. Importantly, *Ae4*⁻/⁻ mice exhibited a significant shift in the relationship between urine titratable acid excretion and plasma HCO₃⁻ concentration (Fig. 3f). This indicates that *Ae4*⁻/⁻ mice lose more HCO₃⁻ into the urine at any given urinary acid secretion rate. Intriguingly, a similar observation has been reported for mice with constitutively increased pendrin activity[28]. Thus, the observed inability of *Ae4*⁻/⁻ mice to reduce pendrin expression resulted in an increased renal loss of HCO₃⁻ in spite of acid-loading. The inability of *Ae4*⁻/⁻ mice to reduce pendrin expression secondary to acid-loading was independent of dietary salt intake (Supplementary Fig. 7). Together, these results demonstrate that AE4 is required for sensing of acidosis and the subsequent rapid downregulation of pendrin activity in β-ICs.

### Regulation of AE4 upon acidosis in pendrin-deficient mice

Interestingly, pendrin deficiency had no impact on AE4 downregulation upon acid-loading (Supplementary Fig. 8a). The AE4 protein abundance was comparably decreased in kidneys of acid-loaded *Pendrin*⁻/⁻ and *Pendrin*⁺/⁺ littermates (Supplementary Fig. 8c). These observations indicate that AE4 regulation does not depend on pendrin activity in β-ICs.

## Discussion

In summary, our study proposes a new, cross-species conserved role for the AE4 (*Slc4a9*) as a crucial element of the basolateral acid-base sensor in the HCO₃⁻ secreting β-ICs. A working model is presented in Fig. 4. In the absence of AE4, neither metabolic alkalosis nor acidosis are perceived resulting in a perturbed physiological regulation of apical pendrin abundance, culminating in life-threatening acid-base disorders. Thus, the central task of the kidneys to ensure acid-base homeostasis is significantly impaired in the absence of AE4. In metabolic alkalosis, AE4 may mechanistically increase the uptake of HCO₃⁻ into β-ICs, thereby driving pendrin transcription, translation, and translocation to the apical membrane, a physiological adaptation essential for the excretion of HCO₃⁻ into urine. Conversely, in metabolic acidosis, AE4-mediated HCO₃⁻ transport into β-ICs is decreased, resulting in a rapid reduction of pendrin abundance and HCO₃⁻ secretion into the urine. Like in mice, pendrin regulation in humans appears to be rapidly regulated by alkali- or acid-challenges[11] suggesting a mechanism of renal adaptation conserved across species that requires AE4. This makes AE4 a novel important player in acid-base balance and a likely pathway involved in renal forms of acid-base disorders.

## Methods
### Antibodies
Primary antibodies used for the study were rabbit anti-AE4 (immunofluorescence [IF] microscopy 1:200, Western Blot [WB] 1:1000, Alpha diagnostics AE41-A), goat anti-AQP-2 (IF 1:500, Santa Cruz sc-9882), guinea pig anti-pendrin (IF 1:500, self-made by C.A. Wagner[29]), rabbit anti-pendrin (WB 1:10,000 self-made by C.A. Wagner[29]), rabbit anti-α-ENaC (WB 1:3000, generous gift by J. Loffing, Institute of Anatomy, University of Zurich, Switzerland[30]), rabbit anti-β-ENaC (WB 1:10,000, StressMarq SPC 404), rabbit anti γ-ENaC (WB 1:3000, StressMarq SPC 405), rabbit anti-NCC (WB 1:5000, Millipore AB3553), sheep anti-NCC Thr60P (WB 1:5000, MRC PPU S995B), rabbit anti-NHE3 (WB 1:10,000, StressMarq SPC 400D), mouse anti-NHE3 Ser552P (WB 1:10,000, Novus NB110-81529 Clone 14D5), rhodamine wheat germ agglutinin (IF 1:400; Vector), and Hoechst (1:1000; Molecular Probes). Secondary antibodies used were either horseradish peroxidase- or fluorescent dye-conjugated, affinity-purified antibodies (anti-goat: IF 1:200, Jackson ImmunoResearch Laboratories; anti-guinea pig: IF 1:200, Jackson ImmunoResearch Laboratories, anti-rabbit: IF 1:200, Jackson ImmunoResearch Laboratories, anti-rabbit: WB 1:5000 to 1:10,000, Dako Cytomation; anti-sheep WB 1:15,000, Thermo Fisher Scientific).

### Genotyping
Mice were genotyped by standard polymerase chain reactions using the following primers: 5′- ggg cag agg aga gag gga gtg-3′ and 5′- tgg tag ctc ctt ccc agg gtg gga-3′ or 5′- cta aag cgc atg ctc cag act gcc-3′ and 5′- tgg tag ctc ctt ccc agg gtg gga-3′. Polymerase chain reaction products were separated by gel electrophoresis on 1.5% agarose gel. The expected wild-type band ran at 900 bp, the mutant band at 350 bp.

### Human and pig organ samples
The human kidney was removed in the setting of a tumor resection, the healthy (tumor free) part of the kidney not needed for pathological diagnosis was used. Patients provided written consent for the use of samples for research. Sample collection was not done by the investigators and all data are anonymized; therefore, they are exempt from IRB approval (§12 HambKHG). Because human samples were anonymized information on age and gender are not available. Pig kidney samples were obtained from freshly sacrificed pigs from a slaughterhouse. 5 mm² kidney pieces were fixed over night at 4 °C in 4% PFA and processed for paraffin blocks.

### Animals and experimental design
All animal protocols were approved by the local authorities (Ministry for Social Affairs, Family, Health and Consumer Protection, Hamburg, Germany, approval number G91/14 and N108/19) and were in accordance with the national and institutional animal care guidelines. For protocols 1–5 male *Slc4a9* knockout (*Ae4*⁻/⁻) and wild-type littermates (*Ae4*⁺/⁺)[18] obtained from heterozygous breeding pairs in the central animal facility (FTH) of the University Medical Center Hamburg-Eppendorf (UKE, Germany) were used. Mice were in a C57BL/6 background and 8–16 weeks old during experimentation. Mice had free access to water and diet, were kept under standardized environmental conditions of temperature and humidity and at a 12-h day/night light cycle. The normal diet contained 0.3% Na⁺ and 0.62% Cl⁻ (Rod16; LasVendi).

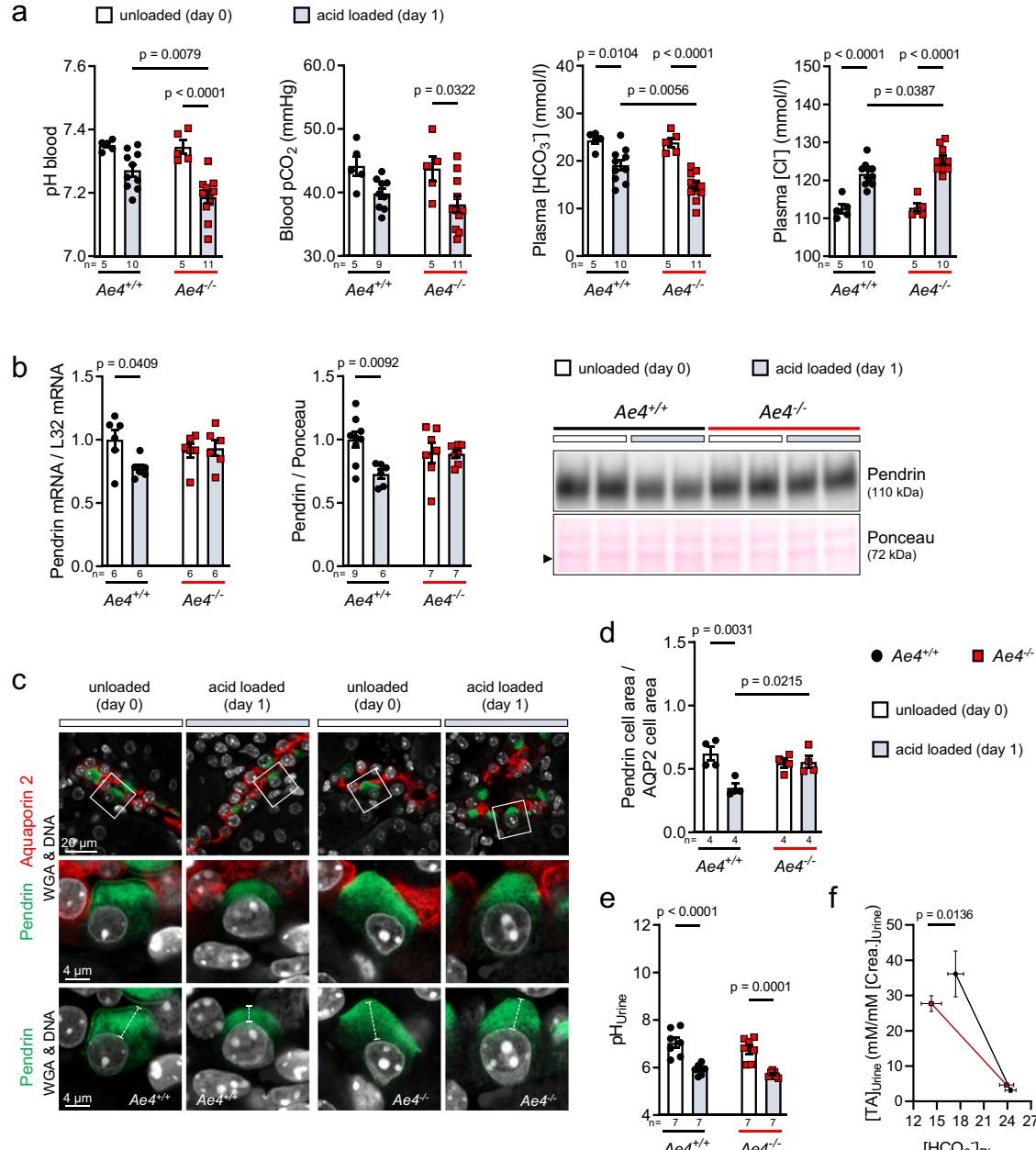

**Fig. 3 | AE4 is essential for downregulation of pendrin in metabolic acidosis.** $Ae4^{-/-}$ and wild-type littermates ($Ae4^{+/+}$) mice were acid-loaded for 1 day under salt-depletion (low salt diet with 280 mM NH$_4$Cl added to the sweetened drinking water). Unloaded (day 0) mice received normal diet. **a** Graphs show blood pH, blood pCO$_2$, plasma [HCO$_3^-$], and plasma [Cl$^-$] of $Ae4^{+/+}$ and $Ae4^{-/-}$ littermates before (day 0) and after 1 day of loading ($n = 5$–11 animals per genotype and day of loading). **b** Representative immunoblots exhibit reduction of pendrin protein abundance in kidney lysates of $Ae4^{+/+}$ but not of $Ae4^{-/-}$ mice after acid-loading. Ponceau red staining served as control for equal loading. Graphs exhibit densitometric quantification of mRNA and protein levels (normalized the levels found in $Ae4^{+/+}$ mice at day 0, $n = 6$–9 animals each genotype and day of loading). **c** Renal immunofluorescence staining of pendrin (green) and AQP2 (red) after 1 day of acid-loading. Boxes mark the area that is magnified, and the length of the dotted lines indicate pendrin "cap size" of unloaded and loaded $Ae4^{+/+}$ and $Ae4^{-/-}$ mice in z-stack confocal images. **d** Quantification of pendrin labeled area / AQP2 labeled area in the renal cortex determined for unloaded (day 0) and acid-loaded (day 1) mice ($n = 4$ animals per genotype and treatment). **e** urine pH ($n = 7$ animals per genotype and treatment) and **f** relationship between urinary titratable acid excretion ([TA]$_{Urine}$ normalized to [creatinine]$_{Urine}$) and plasma [HCO$_3^-$] before and after acid-loading. Note that mice of both genotypes increase urinary acid excretion, but the slope of the [TA]$_{Urine}$ versus [HCO$_3^-$]$_{Plasma}$ curve is significantly higher in $Ae4^{+/+}$ than in $Ae4^{-/-}$ mice ($n = 7$ animals each genotype and treatment, different slope analyzed by simple linear regression). All data are presented as mean ± SEM and in **a**, **b**, **d**, **e** each point or square denotes one animal. In **a**, **b**, **d**, **e** the significance was determined by one-way ANOVA followed by Bonferroni's multiple comparisons test. Source data are provided as a Source Data file.

*Protocol 1 "salt-depletion"*: $Ae4^{-/-}$ and $Ae4^{+/+}$ mice were fed a salt-depleted diet (0.013% Na$^+$; 0.011% Cl$^-$, C1036, Altromin) for 10 days. For analysis of NCC activity, mice were additionally treated with hydrochlorothiazide (1 μg HCTZ (Sigma-Aldrich) per 1 μl 5% DMSO / 95% 0.9% NaCl solution, pH 7–8) at the end of the salt-depleted diet. HCTZ was given by subcutaneous injection (s.c.). Immediately after injection,

mice were placed in metabolic cages for 4 h. Control injection was performed 24 h before with 250 μl 5% DMSO / 95% 0.9% NaCl solution. The HCTZ dose administered was 10 mg/kg body weight.

*Protocol 2 "salt-depletion diet and alkali-loading"*: $Ae4^{-/-}$ and $Ae4^{+/+}$ mice received a salt-depleted diet (C1036, Altromin) for up to 7 days combined with an oral alkali loading (230 mM NaHCO$_3$ in the drinking

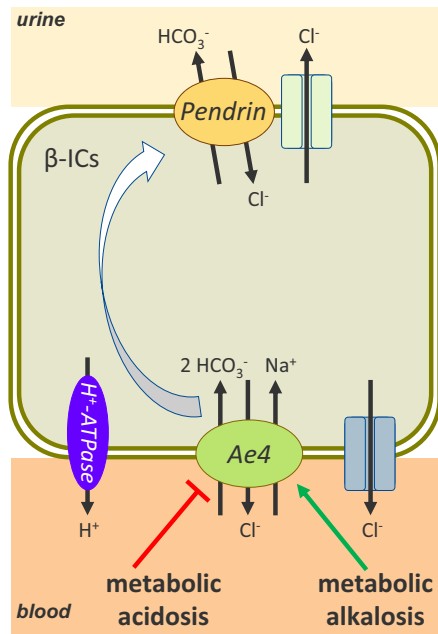

**Fig. 4 | Scheme of β-intercalated cell with proposed AE4 function in renal acid-base homeostasis.** In the absence of AE4 neither metabolic alkalosis nor acidosis is perceived and the physiologic adjustment of pendrin abundance to the acid-base status is perturbed. Mechanistically, uptake of $HCO_3^-$ through AE4 into the β-ICs is assumed to be increased in metabolic alkalosis. This enhanced $HCO_3^-$ transport rate drives pendrin transcription, translation, and translocation to the apical membrane, an adaptation essential for the excretion of $HCO_3^-$ to the urine. In metabolic acidosis, $HCO_3^-$ transport by AE4 is decreased resulting in a rapid reduction of pendrin abundance and $HCO_3^-$ excretion to the urine.

water). For the analysis of dose-dependent effects on acid-base status, the salt-depleted diet was given for 7 days in combination with 80, 130, 180, 240 or 280 mM $NaHCO_3$ in the drinking water.

*Protocol 3 "normal diet and alkali-loading"*: $Ae4^{-/-}$ and $Ae4^{+/+}$ mice on a normal diet (Rod16; LasVendi) received an oral alkali loading (280 mM $NaHCO_3$ in the drinking water) for 1 day (24 h).

*Protocol 4 "salt-depletion diet and acid-loading"*: $Ae4^{-/-}$ and $Ae4^{+/+}$ mice received a salt-depleted diet (C1036, Altromin) for up to 7 days combined with an oral acid loading (280 mM $NH_4Cl$ in sweetened drinking water).

*Protocol 5 "normal diet and acid-loading"*: $Ae4^{-/-}$ and $Ae4^{+/+}$ mice on a normal diet (Rod16; LasVendi) received an oral acid loading (280 mM $NH_4Cl$ in sweetened drinking water) for 1 day (24 h).

*Protocol 6 "salt-depletion diet and acid-loading of Pendrin^{-/-} and Pendrin^{+/+} mice"*: Experiments were performed with male and female *Slc26a4* knockout (*Pendrin^{-/-}*) and wild-type littermates (*Pendrin^{+/+}*) (JAX stock #018424[31]), which were obtained from heterozygous breeding pairs (Aarhus University, Denmark[27]). Mice received a normal diet or a salt-depleted diet (C1036, Altromin) combined with an oral acid loading (280 mM $NH_4Cl$ in sweetened drinking water) for 7 days.

Urine was collected by housing mice in metabolic cages (Techniplast) for 4 h, by puncture of the bladder or as spot urines. At the end of each protocol, blood was collected by penetrating the retro-orbital plexus under deep isoflurane anesthesia. Afterwards, mice were euthanized by cervical dislocation and both kidneys were removed. Tissue was immediately frozen in liquid $N_2$ for further RNA or protein isolation or fixed with 4% paraformaldehyde.

## Immunofluorescence
3 µm paraffin sections were deparaffinized and antigen retrieval was performed by microwave boiling (10 mM citrate buffer pH 6.1) for 30 min at 98 °C. Nonspecific binding was blocked (5% horse serum with 0.05% triton X-100) for 30 min RT. Primary antibody incubations were performed in blocking buffer overnight at 4 °C. After washing in PBS, fluorochrome-labeled donkey secondary antibodies (all from Jackson ImmunoResearch Laboratories) were applied for 30 minutes at RT. Counterstains were performed with rhodamine-wheat germ agglutinin (1:400, Vector) or Hoechst (1:1000, Molecular Probes) together with the secondary antibodies. After washing in PBS, sections were mounted in Fluoromount. Staining was visualized using an LSM800 with Airyscan and the ZenBlue software (Zeiss). To visualize pendrin cap size, z-stack confocal images were obtained by collecting 7 optical sections (0.4 µm) per pendrin cell. Further imaging processing was performed with the ZenBlack software (Zeiss). To quantify the relative abundance of pendrin-positive to AQP2-positive cells three animals per genotype and treatment were used and 20 cortical fields at ×400 magnification were photographed per mouse. All cells with visible nucleus and clear labeling for AQP2 or pendrin were included. To quantify the relative labeling area of pendrin to the labeling area of AQP2, four mice per genotype and treatment were included and 10–12 adjacent cortical fields were photographed per mouse. The stained area per labeling was measured using Image J.

## Plasma volume
Plasma volume was measured in continuously anesthetized mice by means of Evans Blue dilution. 100 µl of a 0.5 mg/ml Evans Blue (Sigma-Aldrich) solution was injected into the retro-orbital plexus with a syringe (30-G needle). After 7–10 min, blood samples were collected by penetrating the retro-orbital plexus into heparinized cups and plasma was separated by centrifugation. Evans Blue absorbance was determined at 620 nm with a Multiskan FC (Thermo Fisher Scientific). Plasma Evans blue concentration was calculated according to a standard curve generated by a serial dilution of Evans Blue in pooled mouse plasma (0–0.1 mg/ml). Plasma volume was calculated using a linear regression model. Blood urea nitrogen was measured by the Institute of Clinical Chemistry and Laboratory Medicine (UKE, Germany) and hematocrit was determined with an ABL 90 series blood gas analyzer (Radiometer).

## Blood gas analysis and plasma electrolytes
Blood gas analysis and measurements of plasma electrolyte concentrations were performed with an ABL 90 series blood gas analyzer (Radiometer).

## Urine analysis
Urine $Na^+$, $K^+$, and $Cl^-$ concentrations were determined using Spotchem EL SE-1520 (Axonlab).

Urinary pH was measured by a micro pH electrode (pH-200, Unisense, Aarhus), urine $[HCO_3^-]$ was determined with an infra-red $CO_2$-sensor based system, urine $[NH_4^+]$ was measured using an Orion™ High-Performance Ammonia Ion-Selective Electrode (Thermo Scientific), and urine [TA] was quantified by titration as previously described in detail[32]. Urinary electrolyte and acid-base concentrations were normalized to the corresponding creatinine concentration, determined according to the Jaffe method, of each probe. Plasma creatinine concentrations were measured by a commercial laboratory (Laboklin, Bad Kissingen, Germany). For each individual mouse, urinary fractional excretion of sodium (FE $Na^+$) was calculated as follows: $([Na^+]_{Urine}*[Creatinine]_{Plasma})/([Na^+]_{Plasma}*[Creatinine]_{Urine})$. Urine osmolality was measured with a freezing point osmometer (Knauer).

## Plasma hormones
For determining plasma renin activity mice were preconditioned to anesthesia handling prior to blood collection. Plasma renin activity was assessed as formation of angiotensin I (Ang I)[33]. The generated Ang I (µg/ml/h, Phoenix Pharmaceuticals Inc.) and plasma aldosterone levels (DRG Instruments) were determined by ELISA following the manufacturer's protocol.

## mRNA quantification

Kidney cortex RNA extraction was performed as described previously[34]. RNA purity and yield were confirmed by spectroscopy (NP50, Implen). mRNA expression levels were quantified by qPCR on a Quant Studio 5 Real-Time-PCR-System (Applied Biosystems, Thermo Fisher Scientific) after RT reactions. All experiments were done using the PowerUp SYBR Green Master Mix (Applied Biosystems, Thermo Fisher Scientific) following the manufacturer's recommendations. Following primers were used: *Slc26a4* 5′-tct acg cca cca agc atg ac-3′ and 5′-gac agc cgt tcg aga cag ag-3′; *Rpl32* 5′-ggt ggc tgc cat ctg ttt tac-3′ and 5′-gat ctg gcc ctt gaa cct tct-3′. Data were analyzed using Quant Studio 5 analysis software. *Slc26a4* (pendrin) expression levels were normalized to *Rpl32* (ribosomal protein L32) levels and expressed as relative mRNA levels.

## Protein quantification

All immunoblots were performed with total kidney lysate from *Ae4*[−/−] and *Ae4*[+/+] littermates or *Pendrin*[−/−] and *Pendrin*[+/+] littermates as follows. Samples were lysed in sucrose solution (250 mM sucrose, 10 mM triethanolamine, 1% protease and 1% phosphatase inhibitor cocktail (Sigma-Aldrich)). The homogenate was centrifuged at $21,000 \times g$ (30 min, 4 °C). The membrane-enriched pellet was resolved with sucrose solution and used for immunoblotting after the determination of protein concentration (Bradford, Bio-Rad, Germany). Samples (10 μg each) were separated on Criterion TGX Precast Gels (Bio-Rad). After protein transfer (TransBlot Turbo Transfer System, Bio-Rad), proteins were visualized by Ponceau red staining. Then nitrocellulose membranes (Bio-Rad) were blocked (5% nonfat milk) before incubation with primary antibodies diluted in solution 1 (Signal Boost Immunoreaction Enhancer Kit, Millipore). Binding was detected by incubation with horseradish peroxidase-coupled secondary antibodies (in solution 2, Signal Boost Immunoreaction Enhancer Kit, Millipore). Protein expression was visualized with ECL SuperSignal (Thermo Fisher Scientific) on a Fusion FX7 EDGE V0.7 Imager (Vilbert Lourmat). Immunoblots were analyzed using ImageJ and Bio 1D (Vilbert Lourmat) software. Ponceau red stainings of the same membrane are shown and were used as loading control and for densitometric normalization. Uncropped scans and blots provided in the Source Data file.

## Measurement of intracellular buffer capacity in β-intercalated cells

Intracellular buffering capacity ($β_i$) was measured in individual β-intercalated cells in perfused cortical collecting ducts from *Ae4*[−/−] and their respective *Ae4*[+/+] littermates. Four experimental groups were studied comparing *Ae4*[−/−] and *Ae4*[+/+] mice on standard chow and on a salt-depleted diet combined with an oral alkali loading (230 mM NaHCO$_3$ in the drinking water) for 1 day. The protocol was similar to that described by D. Good[35]. The tubule lumen was perfused with Cl⁻-free HEPES buffered solution to inhibit anion-dependent base transporters. On the bath side, a Na⁺-free HEPES buffered solution supplemented with 10 nM bafilomycin was used to inhibit any Na⁺-dependent acid base transporter and the V-type H⁺-ATPase. Intracellular base loading was done by adding 2.5 mM trimethylamine to the bath, which rapidly increased pH$_i$. $β_i$ was calculated as $\Delta[HB^+]/\Delta pH_i$, where $\Delta pH_i$ is the increase in pH$_i$ resulting from weak base addition and $\Delta[HB^+]$ is the change in intracellular trimethylammonium concentration, calculated from its pKa (9.8 at 37 °C) and assuming that the concentration of trimethylamine base is equal in intracellular and extracellular fluids at steady state. $β_i$ values in individual cells were plotted as function of pH$_i$. Linear regression of the log-transformed $β_i$ values were used to compare slopes between groups.

## Pendrin activity

Pendrin activity was assessed as previously described[27]. In short, pendrin transport rate was assessed by measuring intracellular pH with fluorescence imaging of isolated perfused cortical collecting duct (CCD) loaded with BCECF-AM (Invitrogen). In between the different groups (*Ae4*[+/+] and *Ae4*[−/−] mice, treated and untreated) the intracellular buffer capacity of β-IC and their resting experimental pH$_i$ values were essentially not different (Supplementary Fig. 4a). This justifies the use of pH$_i$ rate change measurement to report about pendrin function differences. After luminal Cl⁻ was readded to the perfusion solution, which switched back the direction of pendrin-driven Cl⁻/HCO$_3^-$ exchange to HCO$_3^-$ secretion, the intracellular pH (pH$_i$) recovered from alkalization. The initial acidification rate reflects pendrin activity.

## Statistical analysis

Data were analyzed with GraphPad Prism 9 and are shown as mean ± SEM whereby n represents the number of animals (if not otherwise stated). Statistical analysis was performed by two-tailed Student's *t*-test, two-tailed Mann–Whitney test, one-way ANOVA or two-way ANOVA followed by Bonferroni's multiple comparisons tests or Kruskal–Wallis followed by Dunn's multiple comparisons test if not otherwise stated. Exact *p*-values were included in the figures only when statistical analysis revealed significance. $p < 0.05$ was considered statistically significant.

## Reporting summary

Further information on research design is available in the Nature Portfolio Reporting Summary linked to this article.

## Data availability

Key resources such as genotyping, RT-qPCR primers, and antibodies are provided in the method section. All relevant data supporting the key findings of this study are available within the article and its supplementary information files. A reporting summary for this article is available as a Supplementary Information file. Source data are provided with this paper.

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

## Acknowledgements

We thank Stephanie Zielinski and Annett Hasse for excellent technical assistance. This work was supported by DFG grant HU 800/7-2 to C.A.H.

## Author contributions

H.V., C.M.-S., and H.E. designed experiments and wrote the paper. H.V., M.K., L.E., and C.M.-S. performed all experiments except urine acid-base analysis, intracellular buffer capacity, and pendrin activity measurements. P.B. performed urine acid-base analysis. J.L. and S.L.S. analyzed intracellular buffer capacity and pendrin activity. C.A.H. designed and provided *Ae4*$^{-/-}$ mice. C.A.W. provided pendrin antibodies. P.B., J.L., C.A.H., and C.A.W. edited the manuscript.

## Funding

## Competing interests

The authors declare no competing interests.
