## [Peer Review File · Nature Communications]

The AE4 transporter mediates kidney acid-base sensingREVIEWER COMMENTS

Reviewer #1 (Remarks to the Author):

This is an interesting manuscript describing carefully performed experiments characterizing the phenotype of mice lacking AE4. This sodium-dependent chloride-bicarbonate antiporter encoded by *slc4a9* is expressed in the kidney collecting duct, but its function has not been clear. Here, the investigators characterize the phenotype of mice lacking AE4, both at baseline, and when challenged by salt restriction and acid or base. They find that, at baseline, the mice appear normal, and during salt depletion, they only appear to develop a mild elevation in bicarbonate, not evidence for volume depletion. In contrast, when they are challenged by base or acid loads, they become profoundly alkalemic or acidotic. They conclude that there is little evidence for a substantial role for this transporter in sodium (ECF volume) control, but rather that it acts as an acid/base sensor, as mice lacking it are unable to adapt to bicarbonate or acid challenge.

The experiments are all designed and performed well, and most of the results seem clear. There are some minor concerns, which will be discussed below. The larger concerns regard how the experiments are interpreted; the interpretations may be correct, but some questions remain.

The definitive review by Wall and colleagues (reference 20 in this MS) shows a model for beta intercalated cells that include NBCDE (*slc4a9*) in the apical membrane. This model has been disputed (for example by Soleimani and colleagues, and by RNA-Seq data of Knepper on minority cell types), but it does seem to be accepted by many. This probably deserves a more explicit comment (see below).

While the authors imply that the straw man they are attacking is one in which intercalated cells control ECF volume, based on presumed transepithelial transport of sodium and chloride (as above), they here show that alkali loading with salt depletion led to '...severe weight loss...in part due to profound volume depletion, as indicated by increased hematocrit.' In the knockout mice. This deserves a clearer explanation. The figure 3g shows a cell model that does not include an apical sodium entry pathway. This leaves the reader confused about why alkali loading would cause ECF volume depletion in these mice and why the authors conclude that AE4 doesn't play a role in controlling ECF volume.

Additionally, the reason for the absence of a phenotype at baseline in the AE4 KO mice is unclear. Knocking out pendrin causes baseline alkalosis. It seems strange that knocking out AE4 has a large effect during perturbation, but none at baseline (presumably the secreted bicarbonate at baseline is generated by the proton pump). It seems as if this is why the AE4 is described as a 'sensor', but this issue deserves a more complete exposition (also not sure the stoichiometry adds up, as shown).

Minor Comments

page 5 , Line 22 - plasma sodium concentration is usually believed dependent on avp and not renin. Renin is responsible for volume retention.

Page 13 describes Figure 1 as showing, 'The effect of the NCC blocker hydrochlorothiazide (HCTZ, 10 mg/kg) on urinary Na⁺ was not different between both genotypes (mean SEM, n=7-11 per genotype, ****p<0.0001, twoway ANOVA). It is strange to describe an effect as not significant and then show p<0.0001. I understand what is the intention, but this is confusing

In fact, it seems like many of the experiments are really two by two factorial design. In a two way anova, you should have a strain effect, a treatment effect and a treatment by strain effect. To determine whether the response of the control and the knockout to, for example, thiazide is different, one wants to know if the difference is different (in the above, it should not be).

It seems like the two by two experiments would be better analyzed this way, like that in Figure 2A.

Figure 2E: this is the only panel that calls the mice 'WT' and 'KO'. Probably better to stick with Ae+/>+

Figure 3 legend, says, 'Lines indicate pendrin "cap size" of unloaded and loaded Ae4+/>+ and Ae4-/>- mice in z-stack confocal images of pendrin. The figure shows boxes, arrows and dotted lines. All three should be defined. I assume that you mean that the length of the dotted lines indicates the cap size. Why are the arrows there?

and DNA (bottom row)
not clear

Reviewer #2 (Remarks to the Author):

The manuscript by Vitzthum et al examines the role of SLC4A9 (AE4) in regulation the acid base status in response to acid or base loading. The authors confirm the basolateral localization of AE4 B-intercalated cells. These are the cells that express the apical Cl⁻/HCO₃⁻ exchanger pendrin (SLC26A4). In addition, they show that AE4 does not play a significant role in systemic vascular volume regulation (Fig. 1B and 1C). Their more detailed studies demonstrate that mice deficient in AE4 fail to respond appropriately to alkali or acid loading (Figs. 1 and 2). They specifically demonstrate that AE4 is essential for pendrin stimulation in metabolic alkalosis (Fig. 2) and pendrin downregulation in metabolic acidosis (Fig. 3). They conclude that AE4 is essential for sensing changes in acid base status in intercalated cells.

The studies are well designed and executed. The use of various imaging, molecular and functional technics have enhanced the quality of the data. The conclusions are very premature. Many additional studies are needed to ascertain the role of AE4 as a pH sensing mechanism in intercalated cells.

Below are some of the main concerns:

1. An striking deficiency in this paper is the lack of any immunoblots of AE4 in wild type animals in response to any of the experimental maneuvers (salt depletion, metabolic alkalosis, metabolic acidosis) utilized in this article. The authors have included many immunoblots of pendrin, NCC and ENaC in both WT and AE4 deficient mice. Does AE4 show an adaptive regulation similar to pendrin in these experimental maneuvers? In other words, does AE4 show a regulatory pattern similar to the pendrin in metabolic acidosis or metabolic alkalosis? If so, one may conclude that these two transporters (AE4 and pendrin) may show coordinated regulation to acid base disorders.
2. Some of the co-authors of this study, have utilized and are in the possession of pendrin deficient mice. The authors need to perform similar studies in pendrin KO mice and examine the regulation of AE4. Do pendrin deficient mice display a similar effect on AE4 abundance in metabolic acidosis or alkalosis?
3. The pHi studies in Fig. 2E need to be presented in conjunction with the intracellular buffering capacity in both the WT and KO mice in unloaded and loaded states. The rate of recovery from an alkaline intracellular pH without the knowledge on the buffering capacity could be misleading.
4. The authors have the capability to measure the intracellular pH in B intercalated cells (Fig. 2). Have they attempted to measure the AE4 activity using a similar technic?
5. One major missing data is the absence of balance studies. There are no data indicating the daily urine volume or food intake. These are very critical data when it comes to the interpretation of Extended Data in Fig. 3D (Supplemental data?). The AE4 deficient mice show significant weight loss from Day 3 to Day 7 of alkali loading (Extended Data in Fig. 3D). The reason for this weight loss is not clear. Are the animals consuming less food? Or are they drinking less water, or making excessive urine volume. Any of the above possibilities may have direct bearing on the observed outcomes.
6. The data presented in this manuscript may be pointing to the co-regulation of AE4 and pendrin in acid base disorders. This does not necessarily mean that AE4 is the driver of changes in pendrin activity and abundance. That is why performing a similar series of studies in pendrin deficient mice is very critical toward a better understanding of the role for AE4 or pendrin in acid base disturbances.

REVIEWER COMMENTS

Reviewer #1 (Remarks to the Author):

This is an interesting manuscript describing carefully performed experiments characterizing the phenotype of mice lacking AE4. This sodium-dependent chloride-bicarbonate antiporter encoded by *slc4a9* is expressed in the kidney collecting duct, but its function has not been clear. Here, the investigators characterize the phenotype of mice lacking AE4, both at baseline, and when challenged by salt restriction and acid or base. They find that, at baseline, the mice appear normal, and during salt depletion, they only appear to develop a mild elevation in bicarbonate, not evidence for volume depletion. In contrast, when they are challenged by base or acid loads, they become profoundly alkalemic or acidotic. They conclude that there is little evidence for a substantial role for this transporter in sodium (ECF volume) control, but rather that it acts as an acid/base sensor, as mice lacking it are unable to adapt to bicarbonate or acid challenge.

The experiments are all designed and performed well, and most of the results seem clear. There are some minor concerns, which will be discussed below. The larger concerns regard how the experiments are interpreted; the interpretations may be correct, but some questions remain.

The definitive review by Wall and colleagues (reference 20 in this MS) shows a model for beta intercalated cells that include NBCDE (*slc4a9*) in the apical membrane. This model has been disputed (for example by Soleimani and colleagues, and by RNA-Seq data of Knepper on minority cell types), but it does seem to be accepted by many. This probably deserves a more explicit comment (see below).

Author's response. We fully agree with the reviewers' summary of the current controversy regarding the renal function of NDCBE (*Slc4a8*) and its cell specific expression in the renal cortex (Xu et al. , 2018, doi.org/10.1159/000494596 and Chen et al., 2021, 10.1681/ASN.2020101407). However, as our work does not provide any new data regarding the function or localization of NDCBE in the kidney, we would prefer not to comment on this question in the manuscript. Our data do not exclude that β -intercalated cells may participate in sodium reabsorption under certain conditions, but definitely exclude Ae4 as an essential part of sodium reabsorption via β -ICs.

While the authors imply that the straw man they are attacking is one in which intercalated cells control ECF volume, based on presumed transepithelial transport of sodium and chloride (as above), they here show that alkali loading with salt depletion led to '...severe weight loss...in part due to profound volume depletion, as indicated by increased hematocrit.' In the knockout mice. This deserves a clearer explanation. The figure 3g shows a cell model that does not include an apical sodium entry pathway. This leaves the reader confused about why alkali loading would cause ECF volume depletion in these mice and why the authors conclude that AE4 doesn't play a role in controlling ECF volume.

Author's response. To explain the severe weight loss and the volume depletion in the *Ae4*^{-/-} mice upon 7 days of alkali-loading (low salt diet combined with 230 mM NaHCO₃ in the drinking water) we now include additional data in **Supplementary Fig. 3** (please note that FE chloride is only provided for the reviewers) and changed the manuscript accordingly.

The data show that after 7 days of alkali-loading the *Ae4*^{-/-} mice are in a decompensated physiological state. The underlying pathophysiology of the weight loss and volume depletion is complex and multifactorial, with alkalosis and hypokalemia being the major causative forces.

- Supplementary Fig. 3D**, Analysis of drinking volume and food intake during alkali-loading revealed that *Ae4*^{-/-} mice reduced food intake after 4 days of loading. Reduced food intake certainly contributed to the observed weight loss.

- Supplementary Fig. 3E**, Analysis of urine osmolality at day 7 of alkali-loading demonstrate, that *Ae4*^{-/-} mice had a very diluted urine in spite of any increase in drinking volume, indicating that they lost their ability to properly concentrate the urine during alkali-loading. As *Ae4*^{-/-} mice also developed severe hypokalemia (Fig. 1F) upon alkali-challenge, the disrupted water balance might be due to vasopressin resistance and/or altered AQP2 regulation, as even small decreases in plasma [K⁺] perturb AQP2 regulation and can cause diabetes insipidus (Al-Qusairi, L. et al, 2021, doi 10.1152/ajprenal.00655.2020).

- Supplementary Fig. 3E**, Analysis of urinary sodium and chloride excretion (given as fractional excretion (FE) of sodium and chloride) at day 7 of alkali-loading demonstrate that *Ae4*^{-/-} mice exhibited urinary sodium and chloride loss associated with a profound downregulation of pendrin after 7 days of alkali-loading (Fig. 2B) which can explain the urinary chloride loss.

To address the mechanism of renal loss of sodium in AE4, we examined major renal sodium-reabsorbing pathways, namely the sodium-hydrogen exchanger (NHE3) and the α -subunit of the epithelial sodium channel ENaC (α -ENaC). For this purpose, we performed Western blotting with kidneys lysates of unloaded (day 0) and alkali loaded (day 7) *Ae4*^{+/+} and *Ae4*^{-/-} littermates (**Supplementary Fig. 3F**).

NHE3 is inhibited by alkalosis, and consistently protein levels of NHE3 were significantly lower in alkali-loaded (day 7) *Ae4*^{-/-} than in *Ae4*^{+/+} mice. This indicates that the severe metabolic alkalosis (Fig. 1E), which only occurred in *Ae4*^{-/-} mice, led to disruption of the sodium reabsorption via NHE3.

As the alkali-loaded *Ae4*^{-/-} mice also developed hypoaldosteronism (Fig. 1F)

because of hypokalemia, we analysed the protein levels of α -ENaC. Cleaved α -ENaC abundance was reduced in alkali-loaded *Ae4*^{-/-} mice compared with *Ae4*^{+/+} mice.

As ENaC mediates sodium-reabsorption by the renal principal cells, this reduced expression most likely contributed to the observed urinary sodium loss.

In summary, the *Ae4*^{-/-} mice were volume depleted after 7 days of alkali loading as they entered a vicious cycle of aggravated alkalosis, hypoaldosteronism, and hypokalemia (**Supplementary Fig. 4C**), factors affecting sodium and water handling in different nephron segments. Alkalosis affected sodium reabsorption via NHE3 in the proximal nephron, hypoaldosteronism influenced ENaC levels in the principal cells, and hypokalemia altered water reabsorption.

To address this issue, we changed the wording of the manuscript to (page 5-6):

“The severe weight loss was in part due to **reduced food intake (Supplementary Fig. 3D)** and profound volume depletion, as indicated by increased hematocrit, **high plasma renin activity, renal water and sodium loss in the *Ae4*^{-/-} mice (Supplementary Fig. 3E)**. The disrupted renal sodium and water handling was most likely the consequence of the derailed physiological status of the *Ae4*^{-/-} mice after 7 days of alkali-loading. Alkalosis and hypoaldosteronism decreased the abundance of main sodium reabsorption pathways along the nephron (**Supplementary Fig. 3F**), whereas hypokalemia impaired the renal ability to concentrate the urine²².

Additionally, the reason for the absence of a phenotype at baseline in the AE4 KO mice is unclear. Knocking out pendrin causes baseline alkalosis. It seems strange that knocking out AE4 has a large effect during perturbation, but none at baseline (presumably the secreted bicarbonate at baseline is generated by the proton pump). It seems as if this is why the AE4 is described as a ‘sensor’, but this issue deserves a more complete exposition (also not sure the stoichiometry adds up, as shown).

Author’s response. *Ae4*^{-/-} mice show in contrast to *Pendrin*^{-/-} mice some, albeit fixed pendrin activity under all conditions (**Figure 2E**). This preserved basal pendrin activity might explain the absence of acid-base

imbalances in unloaded *Ae4^{-/-}* mice. Even though there is conflicting information regarding acid-base disturbances in *Pendrin^{-/-}* mice under basal conditions (Amlal et al, doi: 10.1152/ajpcell.00033.2010) it has been shown that differences in acid-base status between *Pendrin^{+/+}* und *Pendrin^{-/-}* develop only under chloride-restricted conditions (Verlander et al, doi 10.1152/ajprenal.00474.2005).

Minor Comments

page 5 , Line 22 - plasma sodium concentration is usually believed dependent on avp and not renin. Renin is responsible for volume retention.

Author's response. We totally agree with the reviewer and changed the wording of the manuscript accordingly.

Page 13 describes Figure 1 as showing, 'The effect of the NCC blocker hydrochlorothiazide (HCTZ, 10 mg/kg) on urinary Na⁺ was not different between both genotypes (mean SEM, n=7-11 per genotype, ****p<0.0001, twoway ANOVA). It is strange to describe an effect as not significant and then show p<0.0001. I understand what is the intention, but this is confusing

In fact, it seems like many of the experiments are really two by two factorial design. In a two way anova, you should have a strain effect, a treatment effect and a treatment by strain effect. To determine whether the response of the control and the knockout to, for example, thiazide is different, one wants to know if the difference is different (in the above, it should not be).

It seems like the two by two experiments would be better analyzed this way, like that in Figure 2A.

Author's response. We performed two-way-ANOVA followed by Bonferroni's multiple comparisons test and changed graph and legend accordingly. The *Ae4^{+/+}* and *Ae4^{-/-}* littermates showed increased natriuresis upon HCTZ treatment, but genotypes did not respond differently. To avoid confusion, we have included the "ns information" in the corresponding graphs (Figure 1C and **Supplementary Fig 2A**). The "ns information" is not explicitly shown in any of the other graphs of our manuscript.

Figure 2E: this is the only panel that calls the mice 'WT' and 'KO'. Probably better to stick with *Ae^{+/+}*.

Author's response. Thank you for careful reading! We changed the figure (**Figure 2E**) accordingly.

Figure 3 legend, says, 'Lines indicate pendrin "cap size" of unloaded and loaded Ae4^{+/+} and Ae4^{-/-} mice in z-stack confocal images of pendrin. The figure shows boxes, arrows and dotted lines. All three should be defined. I assume that you mean that the length of the dotted lines indicates the cap size. Why are the arrows there?

and DNA (bottom row)
not clear

Author's response. We removed arrows and changed the legend of figure (**Figure 3C**) as follows:

“C, Renal immunofluorescence staining of pendrin (green) and AQP2 (red) after 1 day of acid-loading. Boxes (upper row) mark the area that is magnified, and the length of the dotted lines indicate pendrin “cap size” of unloaded and loaded Ae4^{+/+} and Ae4^{-/-} mice in z-stack confocal images of pendrin ~~and DNA~~ (bottom row).”

Thank you for careful reading and the very helpful suggestions.

Reviewer #2 (Remarks to the Author):

The manuscript by Vitzthum et al examines the role of SLC4A9 (AE4) in regulation the acid base status in response to acid or base loading. The authors confirm the basolateral localization of AE4 B-intercalated cells. These are the cells that express the apical Cl⁻/HCO₃⁻ exchanger pendrin (SLC26A4). In addition, they show that AE4 does not play a significant role in systemic vascular volume regulation (Fig. 1B and 1C). Their more detailed studies demonstrate that mice deficient in AE4 fail to respond appropriately to alkali or acid loading (Figs. 1 and 2). They specifically demonstrate that AE4 is essential for pendrin stimulation in metabolic alkalosis (Fig. 2) and pendrin downregulation in metabolic acidosis (Fig. 3). They conclude that AE4 is essential for sensing changes in acid base status in intercalated cells.

The studies are well designed and executed. The use of various imaging, molecular and functional technics have enhanced the quality of the data. The conclusions are very premature. Many additional studies are needed to ascertain the role of AE4 as a pH sensing mechanism in intercalated cells.

Below are some of the main concerns:

1. An striking deficiency in this paper is the lack of any immunoblots of AE4 in wild type animals in response to any of the experimental manuevers (salt depletion, metabolic alkalosis, metabolic acidosis) utilized in this article. The authors have included many immunoblots of pendrin, NCC and ENaC in both WT and AE4 deficient mice. Does AE4 show an adaptive regulation similar to pendrin in these experimental maneuvers? In other words, does AE4 show a regulatory pattern similar to the pendrin in metabolic acidosis or metabolic alkalosis? If so, one may conclude that these two transporters (AE4 and pendrin) may show coordinated regulation to acid base disorders.

Author's response. To gain more insight into the regulation of AE4 we performed Western blot experiments with kidney lysates of alkali- and acid-loaded *Ae4*^{+/-} mice and measured AE4 protein abundance as suggested by the reviewer. These data are now shown in

Supplementary Fig.8 A.

In accordance with the findings of Purkerson et al. (2014, 10.1152/ajprenal.00404.2013) we found a significantly reduced AE4 protein level after long-term acid-loading.

Thus, pendrin and AE4 are both downregulated

However, downregulation of AE4 occurred only after 7 days, whereas pendrin was diminished after 1 day already (Fig. 3)

upon acid-loading.

AE4 protein levels were not increased upon base-loading.

2. Some of the co-authors of this study, have utilized and are in the possession of pendrin deficient mice. The authors need to perform similar studies in pendrin KO mice and examine the regulation of AE4. Do pendrin deficient mice display a similar effect on AE4 abundance in metabolic acidosis or alkalosis?

Author's response. To elucidate AE4 regulation in pendrin WT (*Pendrin*^{+/+}) and KO (*Pendrin*^{-/-}) mice, we performed immunostaining and Western blotting (**Supplementary Fig.8 B and C**).

While the basolateral localization of AE4 was not affected by pendrin deficiency (**Supplementary Fig.8B**), unloaded *Pendrin*^{-/-} mice exhibited lower AE4 protein levels compared with *Pendrin*^{+/+} mice (**Supplementary Fig.8C**). Importantly, the observed reduction of AE4 protein levels after 7 days of acid-loading was preserved in *Pendrin*^{-/-} mice. Both, acid-loaded *Pendrin*^{+/+} and *Pendrin*^{-/-} littermates reduced AE4 protein levels by about 40% compared to the corresponding unloaded expression levels (**Supplementary Fig.8C**).

3. The pHi studies in Fig. 2E need to be presented in conjunction with the intracellular buffering capacity in both the WT and KO mice in unloaded and loaded states. The rate of recovery from an alkaline intracellular pH without the knowledge on the buffering capacity could be misleading.

Author's response. We entirely agree with the reviewer. We measured buffer capacity in unloaded and alkali-loaded *Ae4*^{+/+} and *Ae4*^{-/-} littermates. The new data are provided as **Supplementary Fig.4A**, showing that the intracellular buffer capacity was not affected by AE4 deficiency.

Additionally, we have added the following paragraph in the Methods section (page 24-25):

“Measurement of intracellular buffer capacity in β -intercalated cells

We measured the intracellular buffering capacity (β_i) in individual β -intercalated cells in perfused cortical collecting ducts from $Ae4^{-/-}$ and their respective $Ae4^{+/+}$ littermates. Four experimental groups were studied comparing $Ae4^{-/-}$ and $Ae4^{+/+}$ mice on standard chow and on a salt-depleted diet combined with an oral alkali loading (230 mM NaHCO_3 in the drinking water) for 1 day. We used a very similar protocol as described by D. Good⁹. The tubule lumen was perfused with Cl^- -free HEPES buffered solution to inhibit anion-dependent base transporters. On the bath side, a Na^+ -free HEPES buffered solution supplemented with 10 nM bafilomycin was used to inhibit any Na^+ -dependent acid base transporter and the V-type H^+ -ATPase. Intracellular base loading was done by adding 2.5 mM trimethylamine to the bath, which rapidly increased pH_i . β_i was calculated as $\Delta[\text{HB}^+]/\Delta\text{pH}_i$, where ΔpH_i is the increase in pH_i resulting from weak base addition and $\Delta[\text{HB}^+]$ is the change in intracellular trimethylammonium concentration, calculated from its pK_a (9.8 at 37°C) and assuming that the concentration of trimethylamine base is equal in intracellular and extracellular fluids at steady state. β_i values in individual cells were plotted as function of pH_i . Linear regression of the log-transformed β_i values were used to compare slopes between groups.

Pendrin activity

To assess pendrin activity, we performed a maneuver previously described⁵. In short, Pendrin transport rate was assessed by measuring intracellular pH with fluorescence imaging of isolated perfused cortical collecting duct (CCD) loaded with BCECF-AM (Invitrogen). In between the different groups ($Ae4^{+/+}$ and $Ae4^{-/-}$ mice, treated and untreated) the intracellular buffer capacity of β -IC and their resting experimental pH_i values were essentially not different (see **Supplementary Fig. 4A**). This justifies the use of pH_i rate change measurement to report about pendrin function differences. After luminal Cl^- was readded to the perfusion solution, which switched back the direction of pendrin-driven $\text{Cl}^-/\text{HCO}_3^-$ exchange to HCO_3^- secretion, the intracellular pH (pH_i) recovered from alkalization. The initial acidification rate reflects pendrin activity.”

4. The authors have the capability to measure the intracellular pH in B intercalated cells (Fig. 2). Have they attempted to measure the AE4 activity using a similar technic?

Author’s response. We are currently in the process of establishing the technical setup to for AE4 activity measurement, but this still work in progress. Since the protocols have to be established first, the results we have obtained so far are still very premature. We therefore prefer not to present them in this manuscript.

5. One major missing data is the absence of balance studies. There are no data indicating the daily urine volume or food intake. These are very critical data when it comes to the interpretation of Extended Data in Fig. 3D (Supplemental data?). The AE4 deficient mice show significant weight loss from Day 3 to Day 7 of alkali loading (Extended Data in Fig. 3D = **Data 0-7 low/base**). The reason for this weight loss is not clear. Are the animals consuming less food? Or are they drinking less water, or making excessive urine volume. Any of the above possibilities may have direct bearing on the observed outcomes.

Author’s comment: Extended Data Fig.3D corresponds to **Supplementary Fig.3D** in the revised version of the manuscript.

Author’s response. Due to animal law restrictions we are not allowed to house mice for longer than 4 hours in metabolic cages. To overcome this limitation, we measured food intake and water intake of mice housed

in normal cages before and during the 7 days of alkali-loading (low salt diet combined with 230 mM NaHCO₃ in the drinking water). To assess renal water handling we measured urine osmolality at day 7 of alkali loading in *Ae4^{+/+}* and *Ae4^{-/-}* littermates.

To evaluate renal sodium and chloride handling we measured plasma and urinary electrolyte and creatinine concentrations in *Ae4^{+/+}* and *Ae4^{-/-}* littermates after 7 days of alkali loading and determined the fractional excretion by calculating

- FE sodium = $([Na^+]_{Urine} * [Creatinine]_{Plasma}) / ([Na^+]_{Plasma} * [Creatinine]_{Urine})$
- FE chloride = $([Cl^-]_{Urine} * [Creatinine]_{Plasma}) / ([Cl^-]_{Plasma} * [Creatinine]_{Urine})$

We include these new data (drinking volume, food intake, urine osmolality, and FE sodium) in **Supplementary Fig. 3** (please note that FE chloride is only provided for the reviewers) and changed the manuscript accordingly.

The data show that after 7 days of alkali-loading the *Ae4^{-/-}* mice are in a decompensated physiological state. The underlying pathophysiology of the weight loss and volume depletion is complex and multifactorial, with alkalosis and hypokalemia being the major causative forces.

1. **Supplementary Fig. 3D**, Analysis of drinking volume and food intake during alkali-loading revealed that *Ae4^{-/-}* mice reduced food intake after 4 days of loading. Reduced food intake certainly contributed to the observed weight loss.

2. **Supplementary Fig. 3E**, Analysis of urine osmolality at day 7 of alkali-loading demonstrate, that *Ae4^{-/-}* mice had a very diluted urine in spite of any increase in drinking volume, indicating that they lost their ability to properly concentrate the urine during alkali-loading. As *Ae4^{-/-}* mice also developed severe hypokalemia (Fig. 1F) upon alkali-challenge, the disrupted water balance might be due to vasopressin resistance and/or altered AQP2 regulation, as even small decreases in plasma [K⁺] perturb AQP2 regulation and can cause diabetes insipidus (Al-Qusairi, L. et al, 2021, doi 10.1152/ajprenal.00655.2020).

3. **Supplementary Fig. 3E**, Analysis of urinary sodium and chloride excretion (given as fractional excretion (FE) of sodium and chloride) at day 7 of alkali-loading demonstrate that *Ae4^{-/-}* mice exhibited urinary sodium and chloride loss associated with a profound downregulation of pendrin after 7 days of alkali-loading (Fig. 2B) which can explain the urinary chloride loss.

To address the mechanism of renal loss of sodium in AE4, we examined major renal sodium-reabsorbing pathways, namely the sodium-hydrogen exchanger (NHE3) and the α -subunit of the epithelial sodium channel ENaC (α -ENaC). For this purpose, we performed Western blotting with kidneys lysates of unloaded (day 0) and alkali loaded (day 7) *Ae4^{+/+}* and *Ae4^{-/-}* littermates (**Supplementary Fig. 3F**).

NHE3 is inhibited by alkalosis, and consistently protein levels of NHE3 were significantly lower in alkali-loaded (day 7) *Ae4^{-/-}* than in *Ae4^{+/+}* mice. This indicates that the severe metabolic alkalosis (Fig. 1E), which only occurred in *Ae4^{-/-}* mice, led to disruption of the sodium reabsorption via NHE3.

As the alkali-loaded *Ae4^{-/-}* mice also developed hypoaldosteronism (Fig. 1F) because of hypokalemia, we analysed the protein levels of α -ENaC. Cleaved α -ENaC abundance was reduced in alkali-loaded *Ae4^{-/-}* mice compared to *Ae4^{+/+}* mice. As ENaC mediates sodium-reabsorption by the renal principal cells, this reduced expression most likely contributed to the observed urinary sodium loss.

In summary, the *Ae4^{-/-}* mice were volume depleted after 7 days of alkali loading as they entered a vicious cycle of aggravated alkalosis, hypoaldosteronism, and hypokalemia (**Supplementary Fig. 4C**), factors affecting sodium and water handling in different nephron segments. Alkalosis affected sodium reabsorption via NHE3 in the proximal nephron, hypoaldosteronism influenced ENaC levels in the principal cells, and hypokalemia altered water reabsorption. To address this issue, we changed the wording of the manuscript (page 5-6) to:

“The severe weight loss was in part due to **reduced food intake (Supplementary Fig. 3D)** and profound volume depletion, as indicated by increased hematocrit, **high plasma renin activity, renal water and sodium loss in the *Ae4^{-/-}* mice (Supplementary Fig. 3E)**. The disrupted renal sodium and water handling was most likely the consequence of the derailed physiological status of the *Ae4^{-/-}* mice after 7 days of alkali-loading. **Alkalosis and hypoaldosteronism decreased the abundance of main sodium reabsorption pathways along the nephron (Supplementary Fig. 3F)**, whereas hypokalemia impaired the renal ability to concentrate the urine²².”

6. The data presented in this manuscript may be pointing to the co-regulation of AE4 and pendrin in acid base disorders. This does not necessarily mean that AE4 is the driver of changes in pendrin activity and abundance. That is why performing a similar series of studies in pendrin deficient mice is very critical toward a better understanding of the role for AE4 or pendrin in acid base disturbances.

Author's response. To address this important issue, we examined the regulation of AE4 protein abundance in *Pendrin*^{+/+} and *Pendrin*^{-/-} littermates under baseline conditions as well as upon acid loading (new **Supplementary Fig. S8 C and D**). We find that AE4 localization and regulation is not affected by pendrin deficiency. The AE4 protein levels were properly downregulated upon acid-loading in *Pendrin*^{+/+} and *Pendrin*^{-/-}, indicating that AE4 regulation does not depend on pendrin. In contrast, pendrin regulation upon acid-base imbalance is severely impaired in AE4 deficient mice. These data strongly argue against a co-regulation of AE4 and pendrin.

The new data are shown in **Supplementary Fig. S8**, and we have added the following sentence to the manuscript (page 8, lines 16-19):

“Interestingly, pendrin deficiency had no impact on AE4 downregulation upon acid-loading (**Supplementary Fig. 8A**). The AE4 protein abundance was comparably decreased in kidneys of acid-loaded *Pendrin*^{-/-} and *Pendrin*^{+/+} littermates (**Supplementary Fig. 8C**). These observations indicate that AE4 regulation does not depend on pendrin activity in β -ICs.

Thank you for careful reading and the very helpful suggestions.

REVIEWERS' COMMENTS

Reviewer #1 (Remarks to the Author):

The authors have addressed the majority of my concerns.